# Pyroptosis and Its Role in Cervical Cancer

**DOI:** 10.3390/cancers14235764

**Published:** 2022-11-23

**Authors:** Kangchen Li, Jialing Qiu, Jun Pan, Jian-Ping Pan

**Affiliations:** 1Department of Clinical Medicine, Zhejiang University City College School of Medicine, Hangzhou 310015, China; 2Cancer Institute, Second Affiliated Hospital, Zhejiang University School of Medicine, Hangzhou 310009, China; 3Institute of Translational Medicine, Zhejiang University City College, Hangzhou 310015, China

**Keywords:** pyroptosis, cervical cancer, gasdermins, caspases

## Abstract

**Simple Summary:**

Pyroptosis is a kind of programmed cell death, which is characterized by pore formation in the plasma membrane and the release of large amounts of inflammatory mediators. Pyroptosis is associated with diseases such as infectious disease, cardiovascular disease, neurodegeneration and cancer. In this review, the molecular mechanisms of pyroptosis together with its role in the initiation, progression and treatment of cervical cancer are summarized and discussed.

**Abstract:**

Pyroptosis, an inflammatory programmed cell death, is characterized by the caspase-mediated pore formation of plasma membranes and the release of large quantities of inflammatory mediators. In recent years, the morphological characteristics, induction mechanism and action process of pyroptosis have been gradually unraveled. As a malignant tumor with high morbidity and mortality, cervical cancer is seriously harmful to women’s health. It has been found that pyroptosis is closely related to the initiation and development of cervical cancer. In this review the mechanisms of pyroptosis and its role in the initiation, progression and treatment application of cervical cancer are summarized and discussed.

## 1. Introduction

Pyroptosis is a kind of programmed cell death accompanied by a significant inflammatory response. It commonly occurs when cells are infected with pathogens and may form part of an antibacterial response [1]. In 2001, Dr. Cookson named this naturally pro-inflammatory programmed cell death, pyroptosis. It describes a burst of pro-inflammatory chemical signals emanating from dying cells [2]. A noncanonical caspase-11-dependent pathway was first reported in 2013, indicating that lipopolysaccharide (LPS) can cause pyroptosis independently of Toll-like receptor 4 (TLR4) [3]. In 2015, Gasdermin D (GSDMD) was proved to be the key factor for the formation of oligomeric pores in the plasma membrane, resulting in cell rupture, death and then pyroptosis [4].

Pyroptosis helps to resist the invasion of various bacteria, fungi and viruses by enhancing host immune defense responses and removing intracellular replication niches [1]. Pyroptosis usually occurs in immune cells, but it can also take place in certain epithelial cells such as cervical epithelial cells. These processes are initiated within the cell, triggered by danger signals to form a large supramolecular complex called the inflammasome [5]. Pyroptosis has different biological morphology and mechanism from other cell death. For example, in contrast to apoptosis, the inflammasome activates a different type of caspases, such as caspase-1/11 in mice and caspase-1/4/5 in humans [6]. These caspases make contributions to the maturation and activation of pore-forming protein gasdermins and pro-inflammatory cytokines. Pore formation leads to the rupture of the cell membrane, releasing multifarious damage-associated molecular patterns (DAMP) such as DNA, HMGB-1 and ATP, which recruit more immune cells, further perpetuating the inflammatory cascade [7,8]. However, some inflammatory responses do not eradicate the initial stimulus and eventually lead to tissue damage [9]. Pyroptosis is also associated with diseases other than infection, such as cardiovascular disease, neurodegeneration and cancer [10].

Morphologically, all of apoptosis, pyroptosis and necroptosis undergo chromatin condensation. In apoptosis, the nucleus splits into several chromatin corpuscles, while in the process of pyroptosis, however, the nucleus remains intact and gasdermin pores are formed on the cell membrane, resulting in the release of cell contents and the entry of water [11,12,13]. Necroptosis, another kind of programmed necrotic cell death, proceeds through the stepwise activation of receptor-interacting protein kinase-3 (RIPK3) and mixed lineage kinase domain like pseudokinase (MLKL), and shares the common characteristics of pyroptosis that include loss of membrane integrity and inflammation [14,15]. There are complicated cross-talks among apoptosis, pyroptosis and necroptosis. As the central regulator of cell death, caspase-8 acts as the molecular switch to control apoptosis, pyroptosis and necroptosis [16]. Description of the detailed cross-talks among these three kinds of cell death is beyond the scope of this review, but is extensively reviewed elsewhere [14,16,17]. Major similarities and differences among apoptosis, pyroptosis and necroptosis are summarized in Table 1 [12,16,17,18,19,20].

Cervical cancer originates from abnormal cervical cells, often invades neighboring tissues or metastasizes to distant parts of the body. The early clinical manifestations of patients are not obvious; later they may appear as abnormal vaginal bleeding, pelvic pain, sexual intercourse pain and other symptoms. The typical pathological process of cervical cancer is the continuous development of precancerous lesions for 10 to 20 years. The common pathological types of cervical cancer are squamous cell carcinoma and adenocarcinoma, of which the former accounts for 70% [21].

The main cause of cervical cancer is persistent infection of human papillomavirus (HPV) [22]. HPV are non-enveloped DNA viruses with about 170 types. Different types of HPV can cause lesions in different parts of the body such as the cervix, vagina, anus, mouth, tonsil and so on, and increase the risk of cancer [22,23,24]. Patients infected with HPV16 have a high risk of HPV-positive oropharyngeal cancer [23]. HPV6 and HPV11 cause most genital warts and laryngeal papillomas. Sexual contact, as the main mode of transmission of HPV, can further infect the anus and genitalia. Studies show that 99.7% of cervical cancer is closely related to high-risk HPV, of which 70% are HPV16 and HPV18 [25,26]. At present, the main effective way of cervical cancer prevention is HPV vaccination [27]. Diagnosis is usually made by cervical smear screening followed by a biopsy. Further medical images can be used to check whether cancer metastasized. The typical cervical cancer screening is the Pap Smear (PAP) after acetic acid. For symptomatic women, further cervical observation and cervical cytology are required. Earlier screening and diagnosis can reduce the number of cervical cancer cases and deaths through early treatment [21]. Surgery, chemotherapy and radiotherapy are the main treatments of cervical cancer.

Cervical cancer is the fourth leading cause of cancer death in women [28,29]. In the United States (US), there were about 13,000 new cases of cervical cancer and about 4152 deaths in 2019 [30]. The population tends to be younger and cervical cancer is the second most common malignant tumor among women aged 20 to 39 [30]. There are great geographical differences in the prevalence of cervical cancer. Due to the lack of widespread screening techniques, the majority of cases occur in developing countries [31].

In recent years, increasing evidence has shown that there is an association between pyroptosis and cervical cancer. In this review, the molecular mechanisms of pyroptosis together with its role in cervical cancer are summarized and discussed.

## 2. Molecular Mechanisms of Pyroptosis

Pyroptosis is mainly mediated by inflammasomes and the activation of caspase family members including caspase-1, which leads to splicing and polymerization of multifarious Gasdermin family members such as GSDMD, contributing to cell perforation and then cell death [32]. Its activation mechanisms mainly include canonical, noncanonical inflammasome pathways and others.

### 2.1. The Role of Caspase Family

The caspase family plays an important role in regulating homeostasis, cell death and innate immune response. Caspase is expressed as an inactive zymogen consisting of an amino-terminal precursor domain and a carboxy-terminal protease domain which contains catalytic subunits of varying sizes [33]. An important common feature of them is that the active site contains cysteine and specifically breaks the peptide bond after the aspartic acid residue [34].

So far, 14 caspases have been identified in mammals. According to the function of caspases, they can be divided into apoptotic and non-apoptotic classes (caspase-1, -4, -5, -11), and the former is divided into initiator caspases (caspase-2, -8, -9, -10) and executioner caspases (caspase-3, -6, -7) [35,36]. Caspase-1 mediates pyroptosis through recognizing cleavage of GSDMD, which is considered a canonical pathway [37]. With the deepening of the research on cell pyroptosis, a variety of caspases involved in the process of pyroptosis have been found continuously, such as murine caspase-11, and human caspase-4 to recognize cytoplasmic LPS, inducing pyroptosis through the formation of GSDMD pores [38]. Meanwhile, studies have shown that GSDME can be cleaved by caspase-3 to induce pyroptosis, while caspase-6 can activate the related pathways of pyroptosis and apoptosis [33]. Furthermore, caspase-8 also participates in controlling pyroptosis, apoptosis and necroptosis [18].

### 2.2. The Canonical Inflammasome Pathway

Pathogen-associated molecular patterns (PAMPs) present on the surface of toxins, viruses and bacteria, or the DAMPs generated after being stimulated by tissue or cell injury, are identified by intracellular pattern recognition receptors (PRRs) [39,40]. After the recognition, inflammasome complexes are formed, including NLRP1, NLRP2, NLRP3, NLRP4, AIM2, Pyrin, etc. [41]. These inflammasomes activate and cleave pro-caspase-1 through the junction protein apoptosis-associated speck like protein containing a CARD (ASC) to form active caspase-1 [42]. Caspase-1 cleaves self-inhibited GSDMD into an N-terminal domain (GSDMD-N) and C-terminal domain (GSDMD-C) [43]. Subsequently, GSDMD-N binds to phosphatidylinositol and phosphatidyl serine on the inner surface of the cell membrane through membrane-lipid interaction and forms oligomers with an inner diameter 10~20 nm in the lipid bilayer [43]. A large number of pores form a non-selective membrane channel between the medial and lateral sides of the membrane, resulting in the imbalance of ion concentration on both sides of the plasma membrane. The cell swells due to the influx of water while the contents escape through the membrane pores, and the cell eventually dies. In addition, caspase-1 can also activate IL-1β, IL-18 [44] and other small cytoplasmic proteins, and these inflammatory factors are released into the extracellular environment, aggravating the inflammatory response [12] (Figure 1).

### 2.3. The Noncanonical Inflammasome Pathway

After infection with Gram-negative bacteria, caspase-11 in mice or caspase-4/5 in human bodies recognize LPS and become activated to induce pyroptosis. Activated caspase-4, 5, 11 cleaved GSDMD to yield an N-terminal fragment (GSDMD-N, 31 kDa) and a C-terminal fragment (GSDMD-C, 22 kDa). GSDMD-N mediated pyroptosis, but GSDMD-C inhibited the action of GSDMD-N [45]. On the one hand, GSDMD-N forms a wide range of non-selective pores on the cell membrane by dissolving phospholipids, which facilitates the release of IL-18 and IL-1β. On the other hand, GSDMD-N activates NLRP3 inflammatory bodies to activate caspase-1, further inducing pyroptosis. Activated caspase-4, 5, 11 are not involved in the processing of the precursors of IL-1β and IL-18, but indirectly activate the classical pathway of NLRP3-dependent caspase-1 by cutting GSDMD. Caspase-1 shears the pro-IL-1β and pro-IL-18 to form mature IL-1β and IL-18, which are released through membrane channels formed by GSDMD-N, expanding the inflammatory response. It has been found that Caspase-11 also stimulates the pannexin-1 pathway to secrete ATP and K^+^ [46]. ATP activates purinergic receptor P2X7 and opens the P2X7 membrane channel, which further disrupts the integrity of the plasma membrane. K^+^ efflux can activate NLRP3 inflammatory bodies and promote the release of IL-1β, resulting in pyroptosis [47,48,49]. PAMP or DAMP-mediated activation of NF-kB signaling results in elevated NLRP3, pro-IL-1β and pro-IL-18 expression [50,51] (Figure 2).

### 2.4. Other Pathways Inducing Pyroptosis

Caspase-3 has long been regarded as an important executioner of apoptosis [12]. However, in recent years, it has been found that some chemotherapeutic drugs can convert caspase-3-dependent apoptosis into caspase-1-dependent pyroptosis [52,53,54]. Shao and his colleagues found that when knocking out the GSDMD of HeLa cells and replenishing the GSDMD with a caspase-3 cleavage site, HeLa cells can change from apoptosis to pyroptosis [43]. In addition, some studies have found that caspase-3 can influence and activate GSDME to promote the occurrence of pyroptosis. In tumor cell lines with high expression of GSDME, chemotherapeutic drugs like cycloheximide (CHX) can induce caspase-3 activation and cleavage of GSDME [55]. Like GSDMD, GSDME is also a precursor of pore-forming proteins in the GSDM family. Due to the presence of natural cleavage sites, activated caspase-3 can cut specific sites of GSDME, release the active N-terminal domain and perforate the plasma membrane to induce pyroptosis [54,55,56]. In addition, tumor necrosis factor and TLR3, 4 can inhibit an apoptosis inhibitor protein and TGF-β-activated kinase 1 (Tak1), which in turn can activate caspase-8 cleavage of GSDMD and cause pyroptosis [52]. It was found that thioredoxin-interacting protein (TXNIP) is the ligand of NLRP3 and is sensitive to reactive oxygen species (ROS). Some drugs, such as paraquat, can promote the production of ROS, up-regulate the expression of TXNIP and induce the activation of the NLRP3 inflammatory body and the secretion of cytokines, leading to pyroptosis [57].

Recently, it was found for the first time that gasdermin can fulfil a drilling function at non-Asp sites by serine protease granzyme A (GZMA) hydrolysis and proved that pyroptosis induced by cytotoxic lymphocytes was scorched [58]. This discovery rewrites the conclusion that pyroptosis can only be activated by caspase. The serine protease GZMA in cytotoxic lymphocytes (such as CTLs, NK cells) can enter the target cells through perforin and induce pyroptosis by hydrolyzing the Lys229/Lys244 site of GSDMB molecules. GSDMB has tissue-specific expression and is highly expressed in tumor cells originated from epithelial cells of the digestive system. Pyroptosis induced by GSDMB will enhance anti-tumor immunity and become a potential target for the treatment of these tumors [58]. Granzyme B (GZMB) was found to participate in pyroptosis by cleaving GSDME, thereby converting apoptosis to pyroptosis [53] (Figure 3).

## 3. Association between Pyroptosis and Cervical Cancer

### 3.1. Role of Pyroptosis in the Initiation and Progression of Cervical Cancer

#### 3.1.1. HPV Inhibits Pyroptosis of Infected Cells

HPV infection is the major cause of cervical cancer. HPV evades host antiviral immunity through viral genome integration, which can transform infected cervical cells into cancer cells. Some studies have demonstrated that HPV can inhibit the activation of the host adaptive immune gene network that encodes anti-pathogen molecules, chemotaxis and pro-inflammatory factors and proteins involved in antigen presentation. Of which, IL-1β is a key effector molecule in initiating host innate immune response and has a strong immune stimulatory activity. Abnormal release or loss of expression of IL-1β will lead to changes in the local pathological microenvironment of inflammation, which will lead to the loss of normal immune surveillance [59]. As an important secretory factor of pyroptosis, IL-1β is tightly regulated in an inflammatory body/caspase-1-dependent manner at both transcriptional and post-translational levels [60]. Inflammatory bodies usually induce anti-tumor immune response in two ways. One is releasing pro-inflammatory factors through a caspase-1-dependent manner in immunoreactive cells. The other is that NLRP3 inflammatory bodies are activated by ATP to release tumor suppressor factor IL-1β, inducing pyroptosis, thereby removing malignant tumor precursor cells and achieving the effect of anti-tumorigenesis [61]. Therefore, pyroptosis plays an anti-tumor role in the process of tumor formation, while HPV inhibits the progression of pyroptosis by suppressing the expression of IL-1β, IL-18 and inflammasomes, thus promoting tumor progression. It was found that sirtuin 1 (SIRT 1) was over-expressed in HPV-infected cervical cancer cells, and that this increased SIRT 1 inhibits the expression of the AIM 2 inflammasome, hereby suppressing pyroptosis and promoting proliferation of the cancer cells [62]. The HPV E7 protein can inhibit cell pyrogenesis induced by dsDNA transfection. The recruitment of E3 ubiquitin ligase TRIM21 by HPV E7 causes ubiquitination and degradation of IFI16 inflammatory bodies, leading to inhibition of cell pyroptosis and immune surveillance [63]. HPV16 E6 was reversely correlated with the expression of IL-18 [64]. E6 could bind to IL-18 and promote degradation of IL-18 by an ubiquitination pathway, thereby interfering with the local inflammatory process caused by the cascade of downstream effects of IL-18 [64]. Moreover, Matamoros et al. found that local expression of IL-1β and IL-18 was significantly reduced in cervical tissues of patients with cancer compared to that in samples from normal control group, demonstrating that an increase in the risk of progression of pre-neoplastic lesions to cancer was found to be 2.5 and 2.08 times higher in women with lower IL-1β and IL-18 expression, respectively [65]. These findings not only reveal an important immune escape mechanism during HPV infection, but also provide a novel potential target for the antiviral and anti-tumor therapies.

#### 3.1.2. Role of Pyroptosis in the Initiation and Progression of Cervical Cancer: Suppressor or Promotor?

Different pathways of pyroptosis play a role in the occurrence and progression of cervical cancer. NLRP3, the most documented inflammasome, is an inflammasome that exists widely in tumor cells and its expression is regulated by the NF-κB pathway [50]. NLRP3 is also widely expressed in a variety of inflammatory cells, skin keratinocytes and epithelial cells. Current studies demonstrate that the activation of NLRP3 includes a semi-ion channel model, ROS model and a lysosomal rupture model. The ROS model is mainly used to activate the NLRP3 inflammasome to induce pyroptosis of cervical cancer cells [60]. Abdul-Sater et al. infected HeLa cells with Chlamydia trachomatis to stimulate the expression of an inflammasome [66]. They found that Chlamydia trachomatis infection led to the outflow of potassium ions from HeLa cells through its specific potassium channels, thereby stimulating the increase of ROS production. The increased level of ROS induced the assembly of an NLRP3 inflammasome and activated caspase-1. They also found that HeLa cells overexpressing GSDMB exhibit distinct pyroptotic properties [66]. Besides, GZMB released from immune cells cleaves GSDME and promotes pyroptosis of HeLa cells, which is a crucial finding to deduce the connection between GSDM family-mediated cell death and the immune system [53]. GSDME can also be cleaved by activated caspase-3, which removes the intramolecular inhibition of the GSDME-N domain, and GSDME-N has the same effect as GSDMD-N to cause formation of pores in the plasma membrane and pyroptosis [54,56].

Undifferentiated keratinized cells (KCs) in the basal layer of squamous epithelium are the main target cells of HPV infection, and PRR mediated signaling in these cells activates the innate immune system to fight infection [67]. KCs are not only the primary target of HPV infection, but also the part-time immunoreactive cells in the central mucosa. KCs can express and assemble inflammasomes like neutrophils, macrophages and dendritic cells and secrete IL-1β through pyroptosis to resist viral infection [68]. Expression of IL-1β in the cervical cancer group was observably lower compared with that in the normal control group and low-grade squamous intraepithelial lesion group, and women with low IL-1β expression had a higher risk of precancerous lesions progressing to cancer [65]. This is related to HPV-mediated immune evasion [69]. In the precancerous stage of HPV-infected cells, AIM2 plays a tumor suppressor role by activating caspase-1 to promote pyroptosis of tumor cells [62]. Upregulation of microRNA-214 (MiR-214) in cervical cancer patients and cervical cancer cell lines promotes pyroptosis of cervical cancer cells by enhancing NLRP3 expression [63]. These findings demonstrate that pyroptosis plays an inhibitory effect on the initiation and progression of cervical cancer.

However, there are also studies showing that pyroptosis, as a form of cell death accompanied by inflammation, provides a suitable microenvironment for tumor growth. Important factors in the process of pyroptosis, such as inflammasomes, gasdermin proteins and pro-inflammatory immune molecules, are involved in tumorigenesis, invasion and metastasis [70,71]. Being a serine protease and pro-inflammatory factor, GZMB can affect the tumor microenvironment (TME) and promote cancer progression [72]. GZMB has been found to be expressed in urothelial, melanoma and pancreatic cells, which is thought to contribute to cancer invasion [73,74,75]. Moreover, the expression of GZMB in tumor tissues was significantly up-regulated and this expression of GZMB negatively correlated with the survival of patients with cervical cancer [70].

Therefore, pyroptosis in cervical cancer microenvironments might act as a two-edged sword in the progression of the cancer. In the early stage, pyroptosis causes death of tumor cells, which suppresses the progression of cancer. While in the advanced stage, the inflammatory mediators released from pyroptosis might deteriorate inflammation, and pyroptosis of immune cells themselves in the TME compromise their antitumor immune function, thereby promoting mainly cancer progression and metastasis. It requires more preclinical and clinical studies to further evaluate the complicated role of pyroptosis in the progression of cervical cancer.

### 3.2. Pyroptosis in the Diagnosis and Prognosis of Cervical Cancer

Early diagnosis and treatment are crucial to improve the prognosis and survival of patients with cervical cancer. Pyroptosis can regulate cell proliferation, differentiation, invasion and chemotherapeutic drug sensitivity through multiple signal pathways, thus affecting tumor development and is related to the prognosis of patients [76]. Wang et al. integrated the complete set of pyroptosis-related genes (PRGs) and determined the expression changes of PRGs in more than 10,000 cancer cases based on TCGA data [77]. In this study, they found that there were significant changes in the expression of PRGs in different types of cancer, but the expression of PRGs did not always correlate with the survival of patients [77]. Therefore, pyroptosis may play different roles in different cancer types. An immunocorrelation analysis showed that the PRGs, including CHMP4C, TNF and GZMB, were closely related to the tumor immune microenvironment, thereby influencing the progression of cervical cancer [78]. Patients with high TME scores exhibit stronger anti-tumor immune responses, with higher prognosis and survival rates after immunotherapy [79,80]. In addition, progression of pyroptosis can remodel the TME and activate a strong immune cell-mediated anti-tumor response, which in turn plays a strong tumor suppressor effect [81]. These studies demonstrate that when considered as an intrinsic immune mechanism, pyroptosis restrains tumorigenesis and progression, and is beneficial for prognosis and survival.

However, there are also studies showing that pyroptosis is negatively correlated with prognosis. High expression of IL-1β is associated with shorter survival in cervical, gastric and colorectal cancers [82,83]. Earlier studies found that HeLa cells exhibited distinct characteristics of pyroptosis by expressing GSDMB [84]. In the tumor tissues of patients with advanced cervical cancer, the expression of GZMB was significantly up-regulated [70]. The prognostic correlation analysis of cervical squamous cell carcinoma and endocervical adenocarcinoma (CESC) showed that high expression of GZMB was negatively correlated with survival [70]. Therefore, pyroptosis has both positive and negative effects on the development of cervical cancer, and it is worthy of further exploring the value of pyroptosis in the diagnosis and prognosis of patients with cervical cancer.

### 3.3. Application of Pyroptosis in the Treatment of Cervical Cancer

Although there are controversial reports on the role of pyroptosis in the initiation and progression of cervical cancer, it is generally accepted that inducing pyroptosis of the cancer cell is an effective strategy to treat cancer.

#### 3.3.1. Lobaplatin Induces Pyroptosis in Cervical Cancer Cells

As the aforementioned description, pyroptosis mediated by GSDMD-N is typically found in immune cells to respond to inflammatory stimulation, while the one mediated by GSDME-N was lately reported in chemotherapy treated cells [85]. GSDME belongs to the gasdermin superfamily and can be cleaved by caspase-3 [86]. The function of GSDME in the pathogenesis of human tumors has been paid intensive attention. GSDME was considered as a tumor suppressor gene; for example, decreased expression of GSDME mRNA is associated with increased etoposide resistance in melanoma cells [54].

Lobaplatin is a commonly used chemotherapeutic drug and is widely used to treat diverse malignancies, involving cervical cancer, breast cancer, ovarian cancer and so on [87]. Chen et al. demonstrated that lobaplatin induces cervical cancer cell pyroptosis through the caspase-3-GSDME pathway [88]. Lobaplatin activates caspase-3, which subsequently facilitates the cleavage of GSDME and consequently induces pyroptosis. In addition, inhibiting the expression of GSDME significantly suppressed lobaplatin-induced pyroptosis. This research elucidates the interaction between chemotherapy and pyroptosis, emphasizing the critical role of pyroptosis in antitumor chemotherapy.

#### 3.3.2. Tanshinone IIA Induces Pyroptosis in Cervical Cancer Cells

Cisplatin or lobaplatin combined with radiotherapy is the main cytotoxic drug for the treatment of advanced cervical cancer and is considered to be the standard treatment for cervical cancer [89]. However, it was reported that cervical cancer is resistant to chemotherapy and the efficacy of a single drug is often not durable [90].

Tanshinone IIA is a compound extracted from Salvia miltiorrhiza, with multi-biological activities, including antibacterial, anti-inflammatory and antitumor activities [91]. Tanshinone IIA can inhibit malignant proliferation, enhance the death of malignant cells and significantly eliminate them. Tanshinone IIA treatment suppresses the proliferation of cervical cancer cells by repressing the HPV oncogenes and reactivating p53-dependent cancer suppressor genes [92,93].

MiRs are a family of non-coding RNAs that are included in almost all growth and development processes in human bodies [94]. By regulating downstream targets, MiRs can act as tumor suppressor genes or oncogenes. Studies have shown a strong correlation between MiR-145 and inflammation [95]. Zhang et al. revealed that tanshinone IIA inhibited the proliferation and inflammation of HeLa cells, which were associated with the switch between apoptosis and pyroptosis [96]. They found that tanshinone IIA inhibited apoptosis by up-regulating the expression of GSDMD in HeLa cells and the expression of caspase-1 was markedly up-regulated after administration, which promoted the occurrence of pyroptosis, leading to increased levels of IL-18 and IL-1β. At the same time, tanshinone IIA up-regulated the expression of MiR-145 in HeLa cells at the mRNA level and therefore the inhibition of GSDMD expression was relieved [97]. Therefore, regulation of MiR-145 and GSDMD in HeLa cells are tightly related to the anticancer activity of tanshinone IIA, providing potent evidence suggesting that tanshinone IIA is a potential anticancer drug for the therapy of cervical cancer.

#### 3.3.3. Negative Effects of Pyroptosis on Tumor Treatment

It has been shown that the way chemotherapeutic drugs induce cell death is related to the expression level of GSDME. Chemotherapeutic drugs induced apoptosis in cells with low GSDME expression and pyroptosis in cells with high GSDME expression [55]. However, scientists also pointed out that because of methylation, GSDME is under-expressed in most tumor cell lines, while it is generally over-expressed in normal cell lines [98]. Thus, chemotherapy can also lead to pyroptosis of normal cells with high level GSDME, which might be the reasonable ground for chemotherapy-related toxic side effects. This was supported by the study in GSDME knock out mice. Pyroptosis was induced by intraperitoneal injection of cisplatin in wild-type mice, in which severe intestine damage could be observed, while gastrointestinal tissue damages were remarkably reduced in GSDME knockout mice [99].

Pyroptosis is also correlated with the negative effects of radiotherapy. Upon radiation treatment, the AIM2 inflammasome induces caspase-1-mediated pyroptosis of myeloid cells and intestinal epithelial cells, resulting in gastrointestinal injury and toxemia [100]. Therefore, inhibiting the AIM2 inflammasome may alleviate the side effects of radiotherapy. Since radiation also triggers the pyroptosis of bone marrow macrophages mediated by the NLRP3 inflammasome, targeted inhibition of NLRP3 may also be a viable therapeutic strategy [101]. These studies reveal the negative effects of pyroptosis in cancer treatment and provide new insights into chemotherapy and radiotherapy for cancers.

## 4. Conclusions and Future Directions

Pyroptosis is a kind of inflammatory programmed cell death. There are mainly three pathways (canonical, non-canonical and other pathways) to mediate pyroptosis. Complicated cross-talks are present among apoptosis, pyroptosis and necroptosis, and caspase-8 plays a crucial role in the regulation of these three kinds of cell death. Pyroptosis plays different roles in obstetrical and gynecological diseases [102,103]. With respect to the role of pyroptosis in cervical cancer, there are controversial conclusions. Some studies show that pyroptosis plays an anticancer effect, while others demonstrate that pyroptosis negatively correlates with the prognosis of patients with the disease. These findings imply that pyroptosis might play different roles in different phases of cervical cancer. Furthermore, pyroptosis of the cancer cell itself and cells other than cancer cells in the TME also might play different roles in the progression of cervical cancer. Therefore, on the one hand, further studies are required to clarify the specifically regulatory mechanisms of pyroptosis in both cancer cells and other cells in the TME, and particularly, to examine the cross-impact between cancer cells and immune cells in the TME during pyroptosis. On the other hand, more elaborate preclinical and clinical studies are needed to further evaluate the role of pyroptosis in cervical cancer, so that the beneficial and detrimental roles can be exploited and overcome, respectively, in treatment of the patients. Moreover, it would be a very interesting area in the future to study the combined therapy of pyroptosis based intervention with other immunotherapies, such as immune checkpoint inhibition therapy, in cervical cancer.

## Figures and Tables

**Figure 1 cancers-14-05764-f001:**
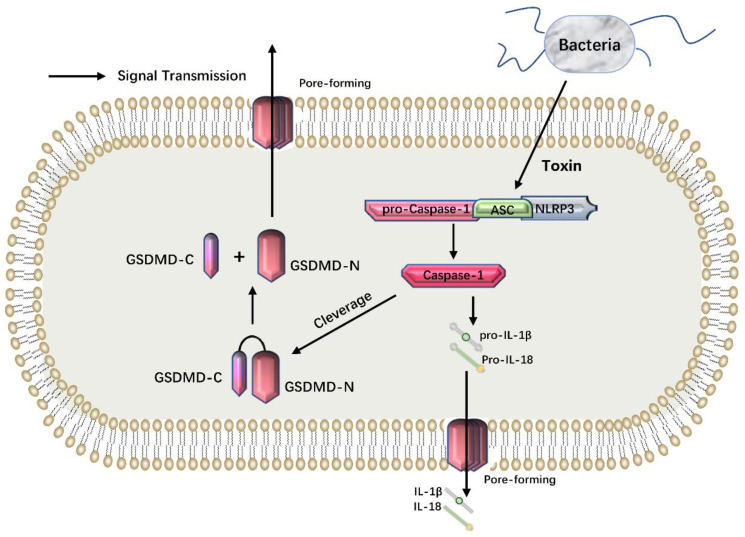
The canonical pyroptosis pathway. PAMPs are recognized by PRRs and form inflammasome complexes, in which NLRP3 is the representative. The inflammasome activates and cleaves pro-caspase-1 into caspase-1 through the adaptor protein ASC, which cleaves GSDMD into an N-terminal domain (GSDMD-N) with perforating activity. GSDMD-N forms non-selective oligomeric pores in the plasma membrane through membrane-lipid interactions. The cells died because of water influx, swelling and rupture. In addition, caspase-1 can also activate inflammatory cytokines like IL-1β and IL-18 which are released into the extracellular fluid via membrane pores and exacerbate the inflammatory response.

**Figure 2 cancers-14-05764-f002:**
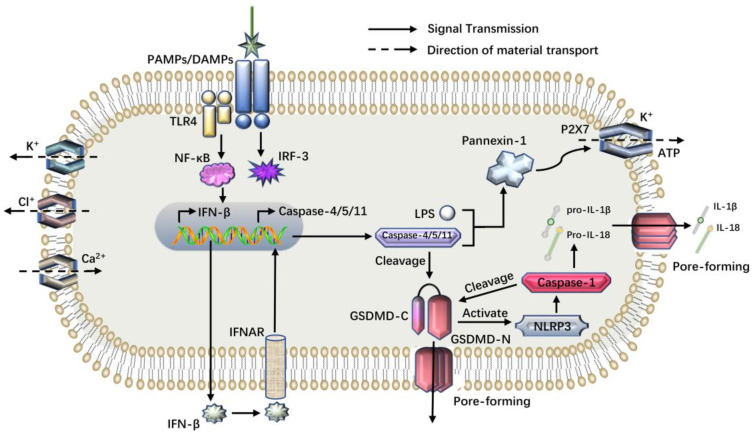
The noncanonical pyroptosis pathway. Endotoxin can activate caspase-11 in mice or caspase-4, 5 in humans to mediate the noncanonical pyroptosis pathway. After activated, caspase-4, 5, 11 cleave GSDMD to generate GSDMD-N, resulting in plasma membrane perforation. At the same time, GSDMD-N activates NLRP3, which activates caspase-1 and initiates the canonical pyroptosis pathway. In addition, by specifically binding to LPS, caspase-11 can facilitate the expression of Pannexin-1. Activated Pannexin-1 activates the P2X7 channel, releasing ATP and K^+^, further damaging the integrity of the cell membrane. Endotoxin can also activate the NF-κB pathway and promote the expression and release of IFN-β.

**Figure 3 cancers-14-05764-f003:**
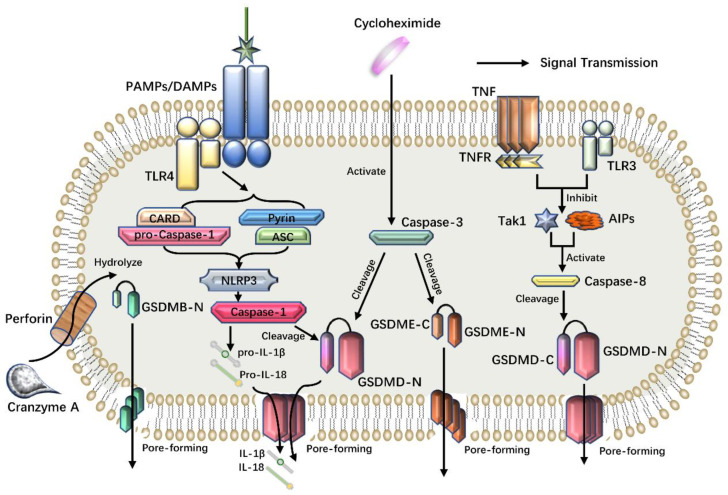
Other pyroptosis pathways. Chemotherapeutic drugs induce the activation of caspase-3, which can cleave GSDME. Once GSDME is cleaved, its N-terminal can also lead to perforation of the plasma membrane, release of inflammatory substances and mediate pyroptosis. In addition, TNF and TLR3 can inhibit AIPs and Tak1, which in turn activates caspase-8 to cleave GSDMD, resulting in pyroptosis. GZMA and GZMB in cytotoxic lymphocytes can enter target cells through perforin and induce pyroptosis by hydrolyzing GSDMB and GSDME, respectively.

**Table 1 cancers-14-05764-t001:** Similarities and differences among apoptosis, pyroptosis and necroptosis.

Characteristics	Apoptosis	Pyroptosis	Necroptosis
Inflammatory	No	Yes	Yes
Chromatin condensation	Yes	Yes	Yes
Nucleus intact	No	Yes	Yes
DNA fragmentation	Yes	Yes	Yes
DNA laddering	Yes	No	No
Membrane integrity	Yes	No	No
Pore formation	No	Yes	Yes
Cell swelling	No	Yes	Yes
Osmotic lysis	No	Yes	Yes
Caspase-1 activation	No	Yes	No
Caspase-3 activation	Yes	Yes	No
Caspase-4/5 activation	No	Yes	No
Caspase-6 activation	Yes	Yes	No
Caspase-7 activation	Yes	Yes	No
Caspase-8 activation	Yes	Yes	No
Caspase-11 activation	No	Yes	No
Cytokines	No	IL-1β, IL-18	TNF-α, IL-6

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
