# Peer review of "Pyroptosis and Its Role in Cervical Cancer"

_cancers, 2022, doi:10.3390/cancers14235764_

Round 1
Reviewer 1 Report
In this review article Li et al. describe some of the molecular mechanisms related to pyroptosis, which is a novel type of programmed cell death. Pyroptosis is the main mechanism for host defense and is crucial for bridging innate and adaptive immunity. The authors describe the relationship between pyroptosis and cervical cancer, which is a public health burden, particularly in developing countries. This review summarizes information about pyroptosis and its relationship with cervical tumorigenesis and shed some light into this relationsip. Since this is a novel nonapoptotic programmed cell death there are many questions that remains to be elucidated.
However, some points should be addressed to improve the paper.
1. In the section “HPV inhibits pyroptosis of infected cells”, the authors should discuss further the role of HPV oncoproteins in the production of anti-inflammatory cytokines, such as IL-18. Since pyroptosis is highly associated with the immune respones, it will be interesting to include this perspective.
2. In the section “Application of pyroptosis in the treatment of cervical cancer, the authors could include a section about the role of pyroptosis in tumor immunological therapy for cervical cancer.
Author Response
- In the section “HPV inhibits pyroptosis of infected cells”, the authors should discuss further the role of HPV oncoproteins in the production of anti-inflammatory cytokines, such as IL-18. Since pyroptosis is highly associated with the immune respones, it will be interesting to include this perspective.
Reply: We appreciate the critical suggestion. We have revised our manuscript by adding the influence of HPV oncoproteins on the expression of IL-18 and AIM 2 inflammasome (lines 244, 245-261 on page 8).
- In the section “Application of pyroptosis in the treatment of cervical cancer, the authors could include a section about the role of pyroptosis in tumor immunological therapy for cervical cancer.
Reply: Thank you very much for your constructive suggestion. To our knowledge, there are some studies reporting the role of pyroptosis in immunotherapy for cancers, such as breast cancer, colorectal cancer, melanoma, etc. According to your suggestion, we searched in PubMed by using the keywords: pyroptosis and immunotherapy of cervical cancer, but there are few documents. Definitely, as you have pointed out, it is a very important and interesting area of research to observe the role of pyroptosis in immunotherapy for cervical cancer. We humbly request that we review this aspect of advances in the future.

Reviewer 2 Report
This manuscript presents interesting data regarding the pathogenetic mechanism of pyroptosis in the initiation, progression, and possible treatment of cervical cancer. The paper is well written, with a very detailed survey on different aspects of pyroptosis in the molecular mechanisms underlying cervical cancer in humans. The use of English is excellent; there are no errors or editing mistakes. The study is fluent and easy to be followed and I have no additional requests on its improvement.
Author Response
This manuscript presents interesting data regarding the pathogenetic mechanism of pyroptosis in the initiation, progression, and possible treatment of cervical cancer. The paper is well written, with a very detailed survey on different aspects of pyroptosis in the molecular mechanisms underlying cervical cancer in humans. The use of English is excellent; there are no errors or editing mistakes. The study is fluent and easy to be followed and I have no additional requests on its improvement.
Reply: We appreciate your positive comments on our manuscript.
